# Feature Analysis and Extraction for Specific Emitter Identification Based on the Signal Generation Mechanisms of Radar Transmitters

**DOI:** 10.3390/s22072616

**Published:** 2022-03-29

**Authors:** Yilin Liu, Shengyong Li, Xiaohong Lin, Hui Gong, Hongke Li

**Affiliations:** Department of Electronic Technology, Naval University of Engineering, Wuhan 430033, China; liuyilin17886@163.com (Y.L.); lsy929@163.com (S.L.); linxiaohong@nudt.edu.cn (X.L.); bigmaner@163.com (H.G.)

**Keywords:** feature analysis and extraction, specific emitter identification, signal generation mechanisms, radar transmitter

## Abstract

In this study, a feature analysis and extraction method was proposed for specific emitter identification based on the signal generation mechanisms of radar transmitters. The generation of radar signals by radar transmitters was analyzed theoretically and experimentally. In the analysis, the main source of unintentional modulation in radar signals was identified, and the frequency stabilization of the solid-state frequency source, the nonlinear characteristics of the radio frequency amplifier chain, and the envelope of the pulse front edge were extracted as features for specific emitter identification. Subsequently, these characteristics were verified through simulation. The results revealed that the features extracted by this method exhibit “fingerprint characteristics” and can be used to identify specific radar emitters.

## 1. Introduction

Specific emitter identification, which is a method used to identify an individual emitter by receiving the emitter’s electromagnetic signals and analyzing the transient fingerprint from the received signals, was first proposed in the 1960s [1]. The complexity of the current electromagnetic environment has increased considerably. Specific emitter identification can be used to precisely detail the distribution of emitters in the battleground and provide support for object identification, electromagnetic interference guidance, and situation awareness.

Generally, specific emitter identification can be categorized into three major parts, namely, signal preprocessing, individual feature extraction, and identification [1,2]. Individual feature extraction is the most critical part in specific emitter identification, which is the transformation from receiving signals to the features that can be used to distinguish individual emitters [3]. The quality of extracted features directly influences the identification effect. Therefore, extracting independent, stable, and reliable features for specific emitter identification is difficult.

Most existing studies on radar emitter recognition are focused on the radar antenna scan pattern, radar waveform, or radar type [4,5,6,7,8,9,10]. Only a few studies have focused on specific emitter identification by improving feature extraction and classification algorithms. For example, deep learning has been adopted to extract highly differentiated features from some primary features, such as pulse envelope and time-frequency pulse images, to distinguish various emitters. However, this method often lacks theoretical support, and extracted features are considerably affected by experimental data [11,12,13]. When the experimental data used are not sufficient or representative, feature extraction is difficult in practice. Therefore, the influence of unintentional modulation on radar signals from each component of radar transmitter has been investigated, and improved methods of feature extraction have been proposed to identify emitters through reliable features.

In 2005, Zhang analyzed the distribution characteristics of phase noise in the frequency synthesizer of the master oscillation power amplification (MOPA) transmitter. This study provided a theoretical basis for extracting individual features of radar signals [2]. In 2008, Xu performed a systematical study on the characteristics of nonlinear components in transmitters. In this study, the Taylor series model was used to describe the amplifier in the transmitter, and the amplitude characteristics of harmonics waves, which are independent of the excitation signal, were investigated [14]. In 2020, Zhang proposed specific emitter identification based on the effect of modulator unintentional modulation on the pulse envelope, which considerably affects a small number of emitters [15]. In these studies, only radar transmitter unintentional modulation was considered. This modulation can effectively distinguish a small number of emitters. However, its performance degrades under conditions of a low SNR or a high number of emitters. In 2021, Peng et al. proposed a nonlinear fingerprint-level radar simulation modeling method, in which nonlinear unintentional modulation of the frequency synthesizer and RF amplifier chain are considered [16]. However, effective extraction of these features is not possible.

To solve the extraction problem, the “fingerprint feature” for specific emitter identification was performed under certain circumstances. Thus, a systematical study on individual features of the MOPA transmitter was conducted by analyzing the influence of unintentional modulation of all components in transmitter on radar signals. Further, some individual features of signals were extracted by various methods to improve the accuracy of specific emitter identification. Finally, the advantages and drawbacks of these extracted features were investigated.

The paper is organized as follows. Section 2 introduces the structure of the MOPA transmitter and the model of a typical radar signal. In Section 3, individual features of each component are extracted and analyzed according to the main sources of unintentional modulation in the transmitter. Section 4 compares the accuracy of identification of features extracted by the simulation experiment. Concluding remarks are presented in Section 5.

## 2. Theoretical Review

### 2.1. Structure of the MOPA Transmitter

The radar transmitter is mainly categorized into the following types: single-stage oscillation and MOPA transmitters. Because the stability, coherence, and tunability of the signal generated by the single-stage oscillating transmitter are not as good as those of the MOPA transmitters, the single-stage oscillating transmitter is seldom used in modern warfare [16]. Therefore, in this study, the MOPA transmitter was investigated.

The composition of the MOPA transmitter, which mainly consists of a solid-state frequency source, radio frequency (RF) amplifier chain, pulse modulator, and high-voltage power supply, is displayed in Figure 1 [17]. The low-power RF signal is generated by the solid-state frequency source and amplified by the RF amplifier chain, and the pulse modulator generates the modulation signal to control the power of RF amplifier chain. When the low-power RF signal is amplified through the RF amplifier chain, it is sent out through the antenna.

### 2.2. Model of the Typical Radar Signal

In ideal circumstances, the signal is sent out through the antenna as follows [17]:(1)$$S(t)=a(t)\cdot A\cos(2\pi f_{c}t)$$
where A is the amplitude of the low-power RF signal generated by the solid-state frequency source, $f_{c}$ is the frequency of the low-power RF signal, and a(t) is the modulated signal generated by the pulse modulator, which can be expressed as follows:(2)$$a(t)=\left\{\begin{array}{c} c\;0+nT\leq t\leq \tau +nT\\ 0\;else \end{array}\right.$$
where *T* is the period of the modulation signal, τ is the pulse width of the modulation signal, and *c* is the gain of the RF amplifier chain. The radar waveform under ideal circumstances is displayed in Figure 2.

## 3. Feature Analysis and Extraction from the MOPA Transmitter

The individual features of the emitter are mainly reflected in the unintentional modulation of the radar signal, and these unintentional modulations are mainly caused by the amplitude noise and phase noise of each component in the radar transmitter [18]. For the MOPA transmitter, the unintentional modulation of the radar signal mainly originates from three parts of the MOPA transmitter, namely, the solid-state frequency source, the RF amplifier chain, and the pulse modulator. In this section, we focused on analyzing and extracting features from the three parts of the MOPA transmitter.

### 3.1. Analysis and Extraction of the Frequency Stabilization of the Solid-State Frequency Source

Because of the advantages of high-spectrum purity and ease of frequency conversion, PLL frequency synthesizers are used as the frequency source for most MOPA transmitters [19]. The composition of the PLL frequency synthesizer is displayed in Figure 3.

The phase discriminator generates the control voltage by comparing it with the reference frequency, which is generated by the divider, and using the voltage-controlled oscillator (VCO) to accurately control the frequency of the output signal, which can be expressed as follows:(3)$$f_{c}=n\cdot f_{r}$$
where $f_{c}$ is the frequency of the output signal generated by the VCO, and $f_{r}$ is the reference frequency generated by the crystal oscillator.

In the PLL frequency synthesizer, unintentional modulation of the output signal is mainly caused by an unstable reference frequency because the actual frequency of the signal generated by the crystal oscillator is not exactly consistent with its nominal because of the deviation in the manufacturing process [20]. Influenced by the deviation in the reference frequency, the frequency of the output signal also exhibits a deviation as follows:(4)$$f_{e}=n\cdot f_{re}$$
where $f_{e}$ is the deviation in the frequency of the output signal, $f_{re}$ is the deviation in the reference frequency. Considering the influence of the deviation in the reference frequency on the output signal, the signal that is sent out through the antenna can be expressed as follows:(5)$$S(t)=a(t)\cdot A\cos(2\pi (f_{c}+n\cdot f_{re})t)$$

The frequency stabilization of the common crystal oscillator can easily reach $10^{-7}$ under general conditions [21]. The test data of the crystal oscillator frequency stability from the two types are presented in Table 1.

To accurately measure the deviation in the frequency of the signal, the accuracy of frequency measurement should be achieved as follows:(6)$$\frac{\Delta f}{f_{c}}<10^{-8}$$
where Δf is the accuracy of frequency measurement.

The method of measuring frequency based on fast Fourier transform (FFT) exhibits considerable robustness and timeliness [22], and the FFT is expressed as follows:(7)$$F(\omega )=\int_{-\infty }^{\infty }S(t)e^{-j\omega t}dt$$

The measurement accuracy of frequency of this method is expressed as follows:(8)$$f=\frac{f_{s}}{L}$$
where $f_{s}$ is the sampling frequency, L is the sampling length. Because of the limitation of the Nyquist sampling theorem, $f_{s}$ should be bigger than $2f_{c}$. Therefore, when the frequency stabilization of the signal reaches $10^{-7}$, the sampling length should reach $2\times 10^{8}$.

Under the aforementioned conditions, the value of the standard deviation of the carrier frequency for multiple pulses is expressed as follows:(9)$${\bar{f}}_{e}=\sqrt{\frac{\sum_{i=1}^{n}\left(f_{ci}-{\bar{f}}_{c}\right)}{n-1}}$$
where n is the number of pulses, $f_{ci}$ is carrier frequency of the *i*th pulse, ${\bar{f}}_{c}$ is the mean value of carrier frequency of all pulses. The value of the frequency stabilization of pulses can be written as:(10)$$K=\frac{{\bar{f}}_{e}}{{\bar{f}}_{c}}$$

The value of the standard deviation of the carrier frequency and the value of frequency stabilization for multiple pulses of one emitter are displayed in Figure 4a,b.

As displayed in the previous figures, the value of the standard deviation of the carrier frequency from the same emitter will change with the carrier frequency, but the value of frequency stabilization will not. Therefore, the value of frequency stabilization can be used as a fingerprint feature to identify emitters. In order to further explore the applicability of frequency stabilization, the values of frequency stabilization for the multiple pulses of three and ten distinct emitters are displayed in Figure 5a,b.

As displayed in the previous figures, when numerous emitters are used, the difference in the frequency stability of some emitters is too low to be distinguished. Therefore, distinguishing emitters only through the frequency stabilization of the carrier frequency is difficult.

### 3.2. Analysis and Extraction of Total Harmonic Distortion of the RF Amplifier Chain

The RF amplifier chain is a complex nonlinear system, which is cascaded by multiple amplifiers [14]. The actual gain curve of the RF amplifier chain is displayed in Figure 6.

In Figure 6, the yellow split line denotes the gain curve of the RF amplifier chain under the ideal conditions. The blue and red lines denote the actual gain curves of the RF amplifier chain that are obtained by measuring two amplifiers of the same type at a frequency of 100 Mhz. Figure 6 reveals that even though the amplifiers are the same type, their gain curves are not exactly identical.

Because of the differences in the amplifier, circuit load, and the stabilization of power supply, the gain curve of the RF amplifier chain in distinct emitters is not exactly identical.

In many studies, the Taylor series model has been used to describe the gain of the RF amplifier chain [23,24]. When the input signal is expressed as s(t)=Acos(2πft), the output signal of RF amplifier chain can be expressed as follows:(11)$$S(t)=\sum_{n=0}^{\infty }b_{n}s^{n}(t)$$

Assuming that the fourth-order and higher-order harmonics generated by nonlinearity are ignored, the output signal of the RF amplifier chain can be expressed using a third-order Taylor series as follows:(12)$$\begin{array}{ll} S(t) & =b_{1}A\cos(2\pi ft)+b_{2}A^{2}\cos^{2}(2\pi ft)\\ & +b_{3}A^{3}\cos^{3}(2\pi ft) \end{array}$$

Through the trigexpand, the formula can be expressed as follows:(13)$$\begin{array}{ll} S(t)= & \frac{1}{2}b_{2}A^{2}+(b_{1}A+\frac{3}{4}b_{3}A^{3})\cdot \cos(2\pi ft)+\\ & \frac{1}{2}b_{2}A^{2}\cos(4\pi ft)+\frac{1}{4}b_{3}A^{3}\cos(6\pi ft) \end{array}$$

According to the aforementioned formula, when the nonlinear characteristics of the RF amplifier chain in various emitters are not exactly same, the amplitude of harmonic of output signals differs, as displayed in Figure 7.

As displayed in the aforementioned figures, various gain curves of the RF amplifier chain result in different harmonics amplitudes.

To extract the total harmonic distortion (THD) of the RF amplifier chain, (11) can be expressed as follows:(14)$$\begin{array}{ll} S(t) & =A_{0}+A_{1}\cdot \cos(2\pi ft)+A_{2}\cos(4\pi ft)\\ & +A_{3}\cos(6\pi ft) \end{array}$$

The amplitude of the signal carrier, second harmonic, and third harmonic $A_{1},A_{2},A_{3}$ can be measured through the FFT on the output signal. Next, the ratio of the nonlinear coefficient, which is independent of the excitation signal, can be expressed as follows:(15)$$\begin{array}{ll} k_{1} & =\frac{b_{1}{}^{2}}{b_{2}}=\frac{{\left[\left(b_{1}A+\frac{3}{4}b_{3}A^{3})-3\cdot (\frac{1}{4}b_{3}A^{3}\right)\right]}^{2}}{\sqrt{2\cdot (\frac{1}{2}b_{2}A^{2})}}\\ & =\frac{{\left(A_{1}-3A_{3}\right)}^{2}}{\sqrt{2A_{2}}} \end{array}$$
(16)$$\begin{array}{ll} k_{2} & =\frac{b_{1}{}^{3}}{b_{3}}=\frac{{\left[\left(b_{1}A+\frac{3}{4}b_{3}A^{3})-3\cdot (\frac{1}{4}b_{3}A^{3}\right)\right]}^{3}}{4\cdot \frac{1}{4}b_{3}A^{3}}\\ & =\frac{{\left(A_{1}-3A_{3}\right)}^{3}}{4\cdot A_{3}} \end{array}$$

According to the aforementioned formula, the coordinate distribution of $k_{1},k_{2}$ is displayed in Figure 8a–c.

Figure 8a–c represents the coordinate distributions of $k_{1},k_{2}$ from two distinct emitters under 30, 20, and 10 dB SNR. The nonlinear coefficient can be used for specific emitter identification, but the accuracy is not high when the number of emitters is large or the SNR is low, as displayed in Figure 9 and Figure 10a–c.

Figure 9 is the gain curve of the RF amplifier chain in six distinct emitters, and Figure 10a–c denotes the coordinate distributions of $k_{1},k_{2}$ under the condition of 30, 20, and 10 dB SNR.

The difference in the SNR considerably influences analysis and extraction of the THD of the RF amplifier chain.

To reduce the influence of environmental noise, some pulses were considered as a group of data, and the mean values of $k_{1},k_{2}$ for each group of data were calculated. The coordinate distribution of mean values of $k_{1},k_{2}$ are displayed in Figure 11a,b.

Figure 11a shows the coordinate distribution of mean values of $k_{1},k_{2}$, which has 10 pulses in each group, and Figure 11b has 20 pulses in each group. These figures reveal that the mean value statistics method can reduce the influence of environmental noise to a certain extent.

### 3.3. Analysis and Extraction of the Envelope Characteristic of the Pulse Front Edge

In ideal circumstances, the modulated signal is a rectangular wave. However, the influence of circuit parameters affects the real waveform of the modulated signal, as displayed in Figure 12. The differences in the power supply, energy storage element, and pulse transformer result in the differences in the pulse front edge, pulse top, and pulse back edge of the output signals from different emitters.

As shown in Figure 13, the envelope of the pulse top and back edge is severely affected by the multipath effect and noise from the environment [25]; therefore, this study only focused on the pulse front edge of the signals. During the time of the pulse front edge, the equivalent circuit of the pulse modulator is displayed in Figure 14.

According to the circuit in the Figure 14, the differential equation of this circuit can be expressed as follows:(17)$$\left\{\begin{array}{c} E_{p}=u_{L}+iZ_{0}+L_{L}\frac{di}{dt}\\ i=C_{s}\frac{du}{dt}+\frac{u_{L}}{R_{L}} \end{array}\right.$$
where $E_{p}$ is the discharge voltage of the energy storage element, and $Z_{0}$ is the impedance of the energy storage element. By solving the aforementioned differential equations, the pulse front edge can be expressed as follows:(18)$$\tilde{a}(t)=u\left\{1-e^{-\alpha t}[ch(Kt)+\frac{\alpha }{K}sh(Kt)]\right\}$$
where ch(·) is the hyperbolic cosine function, sh(·) is the hyperbolic sine function, and u is the steady-state voltage of the circuit, which can be expressed as follows:(19)$$u=E_{p}\frac{R_{L}}{R_{L}+Z_{0}}$$

α and K are both constants, and can be expressed as follows:(20)$$\alpha =\frac{1}{2}\left(\frac{Z_{0}}{L_{L}}+\frac{1}{C_{s}R_{L}}\right)$$
(21)$$K=\sqrt{\alpha^{2}-\frac{1}{L_{L}C_{S}}\left(1+\frac{Z_{0}}{R_{L}}\right)}$$

The above formula reveals that the envelope characteristics of the pulse front edge are mainly affected by $E_{p}$, $Z_{0}$, $R_{L}$, $L_{L}$, and $C_{s}$.

Consider $E_{p}$ = 12,000 v, $Z_{0}$ = 50  Ω, $R_{L}$ = 10,000 Ω, $L_{L}$ = 100 mH, $C_{s}$ = 900 pf in (18) to obtain the front edge of the simulation modulation signal, as displayed in Figure 15.

Extracting and analyzing the characteristics of so many components in the pulse modulator is difficult. Therefore, the Hilbert transform was used to directly extract the envelope of the pulse front edge in this study. The Hilbert transform can be expressed as follows [27]:(22)$${\hat{S}}_{1}(t)=H[S_{1}(t)]=\frac{1}{\pi }\int_{-\infty }^{\infty }\frac{S_{1}(\tau )}{t-\tau }d\tau$$
where $S_{1}(t)$ is the front edge of the output signal, and the envelope of the pulse front edge extracted by Hilbert transform can be expressed as follows:(23)$$\begin{array}{ll} {\tilde{S}}_{1}(t) & =S_{1}(t)+j{\hat{S}}_{1}(t)\\ & =\tilde{a}(t).(b_{1}A+\frac{3}{4}b_{3}A^{3})\\ & =A_{1}\cdot \tilde{a}(t) \end{array}$$

The envelope of the pulse front edge $\tilde{a}(t)$, which is independent of $A_{1}$, can be obtained through normalization. The envelope of the pulse front edge from a single pulse is displayed in Figure 16; the blue line denotes the front edge of the output signal and the orange line denotes the envelope of the pulse front edge.

The above figure reveals that the envelope is not a smooth curve because of the nonlinear amplification from the RF amplifier chain and the noise from the environment. Therefore, filtering the envelope is necessary. The envelope after filtering by downsampling is displayed in Figure 17.

The envelope of the pulse front edge of the single pulse is easily affected by environmental noise and becomes indistinguishable. The envelope characteristics of the pulse front edge of the single pulse from three distinct emitters under conditions of 30, 20, and 10 dB SNR are displayed in Figure 18a–c.

To reduce the influence of environmental noise on the envelope characteristics of the pulse front edge, 100 pulses were considered as a group of data to calculate the mean value of their envelope characteristics; the mean curves of the envelope are displayed in Figure 19. Figure 19a–c displays the mean curves of the envelope from three distinct emitters under conditions of 30, 20, and 10 dB SNR.

The figures reveal the following:(1)The mean curves of envelope characteristics exhibit excellent robustness.(2)Because of the many dimensions of the mean curves of the envelope characteristics, they can be distinguished by some algorithms when the number of emitters is large.

## 4. Analysis of the Simulation Experiment

Section 3 reveals that the extracted features can distinguish emitters in certain conditions. However, in general, the accuracy of specific emitter identification is not satisfactory when only a part of the radar transmitter unintentional modulation is considered. In this section, based on a simulation experiment, we mainly focus on the performance of specific emitter identification that utilizes all the features in Section 3.

### 4.1. Experimental Design

In general, multiple continuous pulses collected by receivers in the same direction and at the same frequency all originate from one emitter. Therefore, features can be extracted from a group of pulses that originate from the same emitter. The experimental process is displayed in Figure 20.

First, the method in Section 3 was used to extract the frequency stabilization of the solid-state frequency source, the nonlinear characteristics of the RF amplifier chain, and the envelope characteristics of the pulse front edge. Next, these features were regarded as a group of feature vectors. Finally, feature vectors were identified using the random forest algorithm.

To research the effect of specific emitter identification based on the aforementioned features under the distinct SNR conditions, 10 emitters were simulated in this study. The characteristic parameters of components in the MOPA transmitter are listed in Table 2. Each emitter generated 10,000 pulses, of which 80% pulses were used as training data, and 20% pulses were used as training data.

### 4.2. Experiment Result

To research the influence of various features being extracted for specific emitter identification, we categorized 8000 pulses into 800 groups of 100 pulses, and extracted all features from each group. Next, each of the features was used to train a random forest classifier, and tested with 200 groups of 100 pulses. The experimental results are displayed in Table 3.

As presented in Table 3, the extracted features can effectively identify emitters. The accuracy of identification, in which all features are extracted, is superior to that using features only from a single component. The low accuracy of frequency stabilization is mainly attributed to the low number of measured pulses, which makes it difficult to distinguish some emitters with similar frequency stabilization.

To research the influence of the number of pulses in each group on specific emitter identification, we divided 10,000 pulses into 1000 groups of 10 pulses, 500 groups of 20 pulses, and 200 groups of 50 pulses. The experimental results are presented in Table 4.

As presented in Table 4, the use of the statistical average can effectively reduce the influence of Gaussian noise on features. However, if too many pulses are used to calculate the mean value of the features, fewer data are required to train the classifier. Therefore, the recognition ability of the classifier is reduced, and the accuracy of identification for specific emitter identification is reduced.

Finally, the proposed method was compared with the method of specific emitter identification, which is based on the combination of conventional features and deep learning through the experiment. The experimental results are presented in Table 5.

Table 4 reveals that the accuracy of identification through the method proposed in this paper is superior to that of specific emitter identification based on the combination of conventional features and deep learning, especially in the case of low SNR.

## 5. Conclusions

In this study, we mainly focused on the method of extracting and analyzing features for specific emitter identification based on the signal generation mechanisms of the MOPA transmitter. First, the main source of unintentional modulation was identified by analyzing the process of radar signal generation. Next, the frequency stabilization, nonlinear coefficients, and envelope of the pulse front edge were extracted as features for specific emitter identification from the PLL frequency synthesizer, RF amplifier chain, and pulse modulator, which are the main sources of unintentional modulation. This was achieved using FFT, Hilbert transform, and other methods. Finally, the effectiveness of features extracted for specific emitter identification was verified through simulation experiments and compared with the method of specific emitter identification, which is based on the combination of conventional features and deep learning. Simulation results revealed that the proposed method outperformed the method based on the combination of conventional features and deep learning in low SNR conditions.

## Figures and Tables

**Figure 1 sensors-22-02616-f001:**
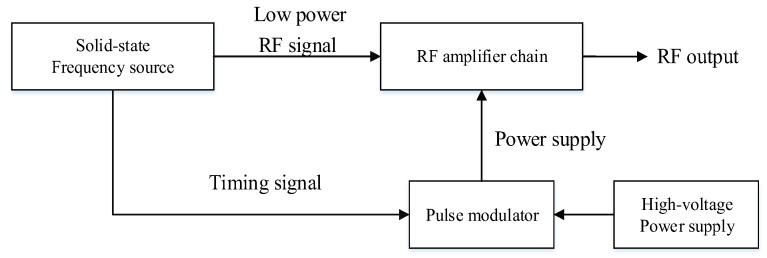
Composition of the MOPA transmitter, where the low-power RF signal is generated by the solid-state frequency source and amplified by the RF amplifier chain.

**Figure 2 sensors-22-02616-f002:**
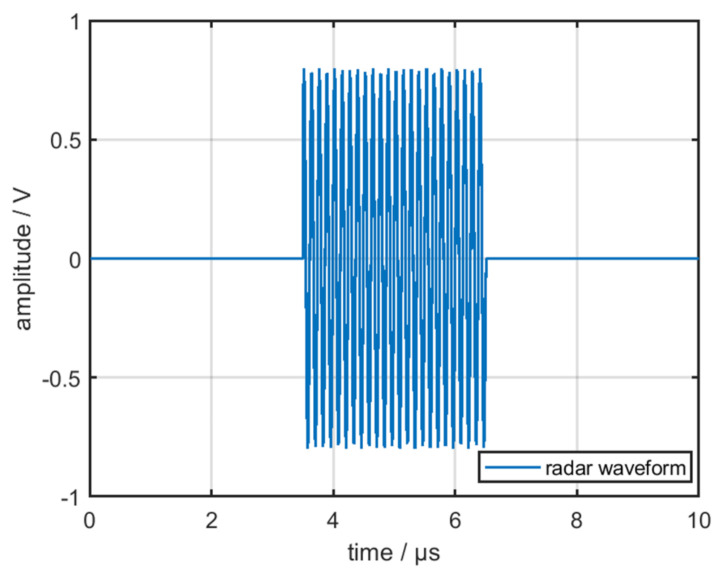
Radar waveform under ideal circumstances.

**Figure 3 sensors-22-02616-f003:**
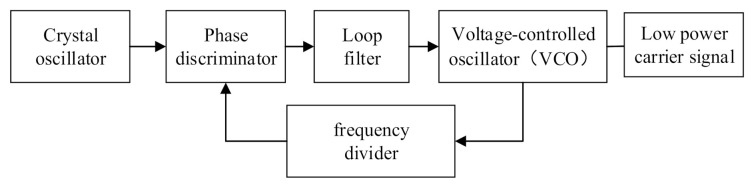
Composition of the PLL frequency synthesizer, where the reference frequency is generated by the crystal oscillator, and the low-power carrier signal is generated by the VCO; the frequency of the low-power carrier signal is controlled by the phase discriminator.

**Figure 4 sensors-22-02616-f004:**
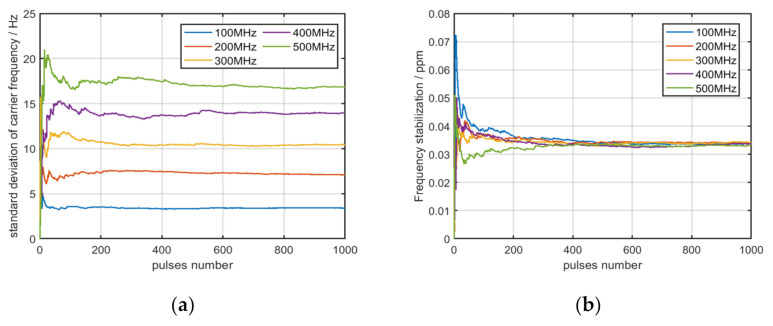
(**a**) The value of standard deviation at five different carrier frequencies; (**b**) the value of frequency stabilization at five different carrier frequencies.

**Figure 5 sensors-22-02616-f005:**
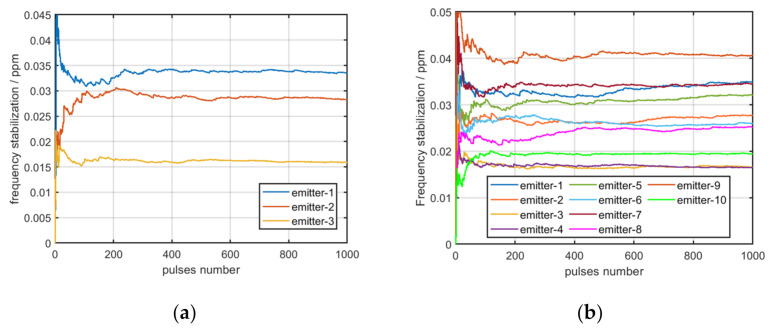
(**a**) Frequency stabilization of three emitters; (**b**) frequency stabilization of ten distinct emitters; the x-axis represents the number of pulses used to calculate the standard deviation of the frequency, and the y-axis represents the value of frequency stabilization.

**Figure 6 sensors-22-02616-f006:**
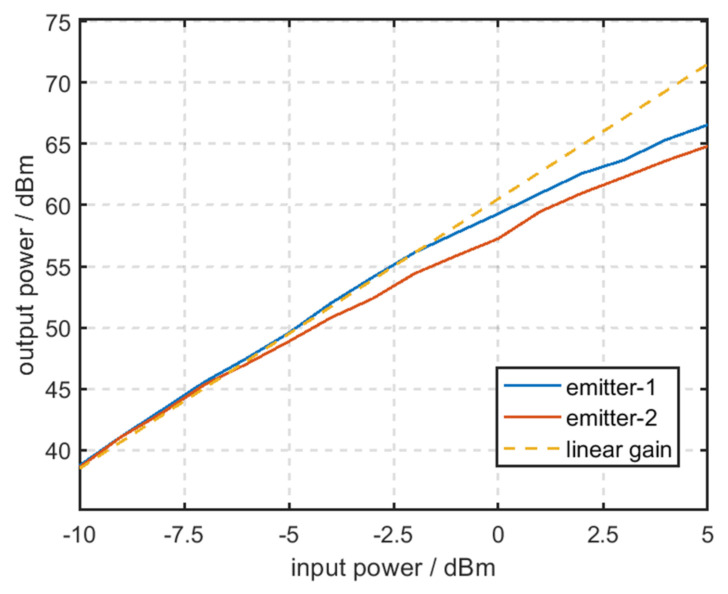
Actual gain curve of the RF amplifier chain.

**Figure 7 sensors-22-02616-f007:**
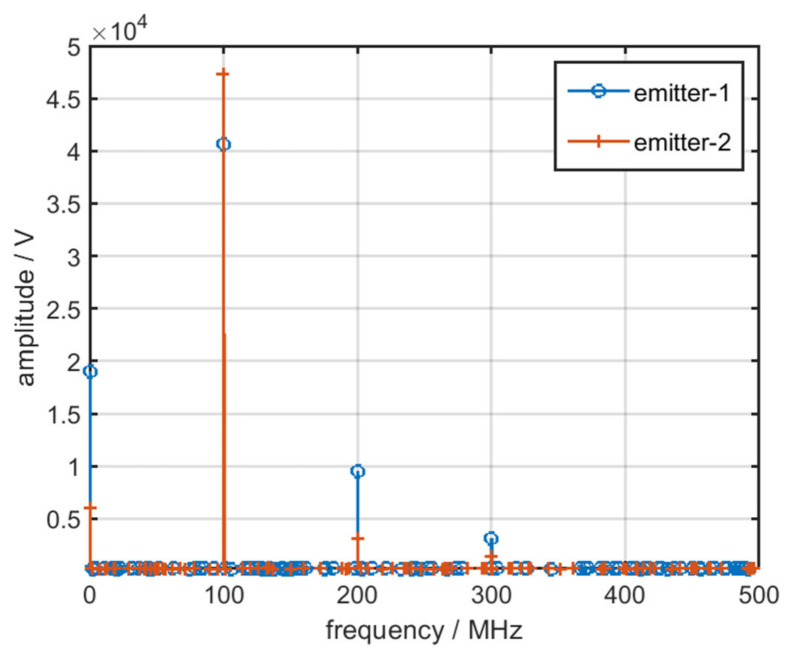
Frequency spectrum of two distinct emitters.

**Figure 8 sensors-22-02616-f008:**
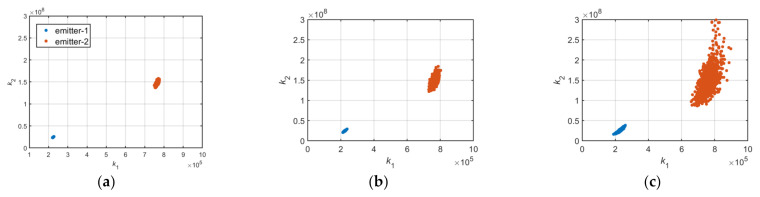
(**a**) $k_{1},k_{2}$ of two distinct emitters at 30 dB SNR; (**b**) $k_{1},k_{2}$ of two emitters at 20 dB SNR; (**c**) $k_{1},k_{2}$ of two emitters at 10 dB SNR.

**Figure 9 sensors-22-02616-f009:**
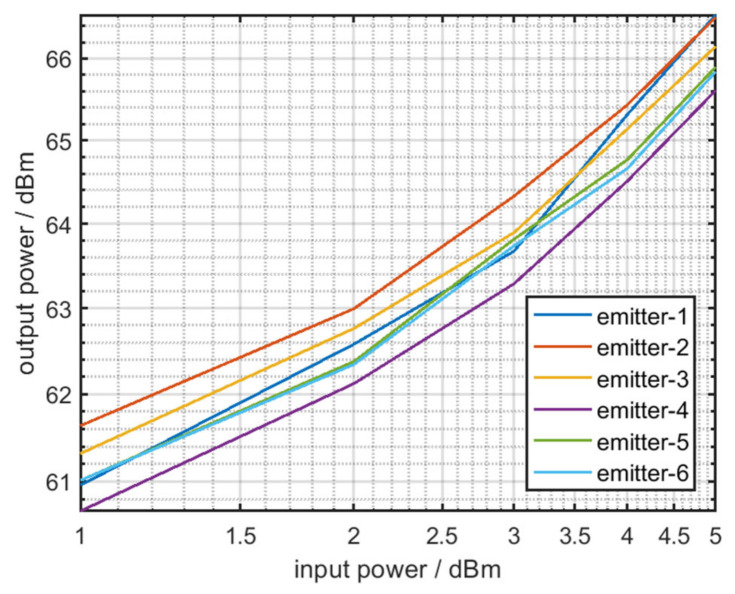
Gain curve of the RF amplifier chain of six distinct emitters.

**Figure 10 sensors-22-02616-f010:**
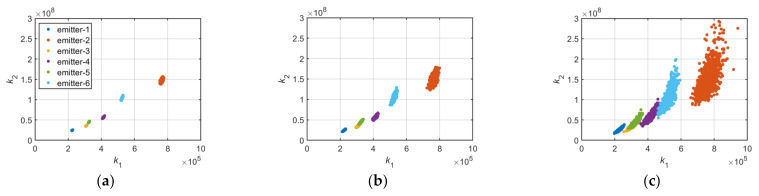
(**a**) $k_{1},k_{2}$ of six distinct emitters at 30 dB SNR; (**b**) $k_{1},k_{2}$ of six distinct emitters at 20 dB SNR; (**c**) $k_{1},k_{2}$ of six distinct emitters at 10 dB SNR.

**Figure 11 sensors-22-02616-f011:**
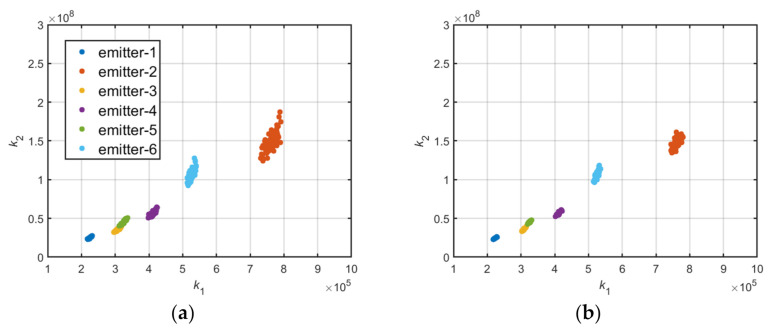
(**a**) Mean value of $k_{1},k_{2}$ of six distinct emitters at 10 dB SNR (10 pulses in each group); (**b**) mean value of $k_{1},k_{2}$ of six distinct emitters at 10 dB SNR (20 pulses in each group).

**Figure 12 sensors-22-02616-f012:**
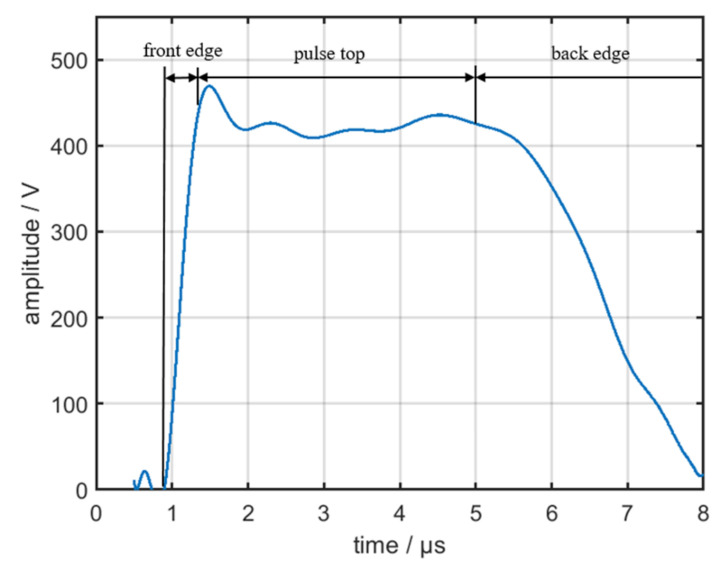
Real waveform of the modulated signal.

**Figure 13 sensors-22-02616-f013:**
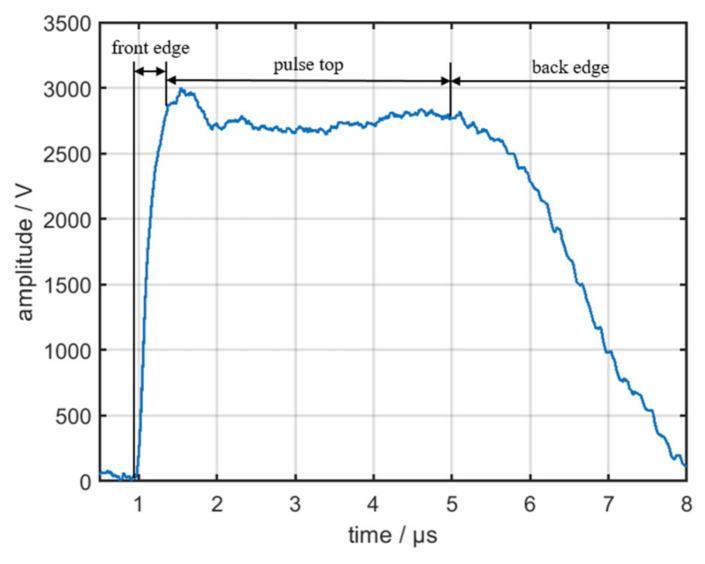
Envelope of the radar signal.

**Figure 14 sensors-22-02616-f014:**
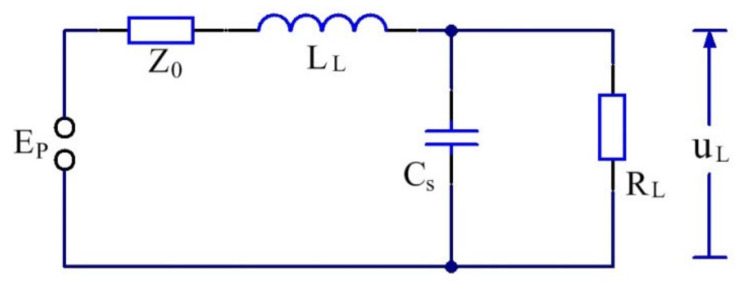
Equivalent circuit of the pulse modulator at the time of the pulse front edge [26], where $E_{p}$ is the discharge voltage of energy storage element,
$Z_{0}$ is the impedance of energy storage element, $L_{L}$ is the excitation inductance of pulse transformer, $C_{s}$ is the distributed capacitance of pulse transformer, and $R_{L}$ is the equivalent impedance of the pulse transformer.

**Figure 15 sensors-22-02616-f015:**
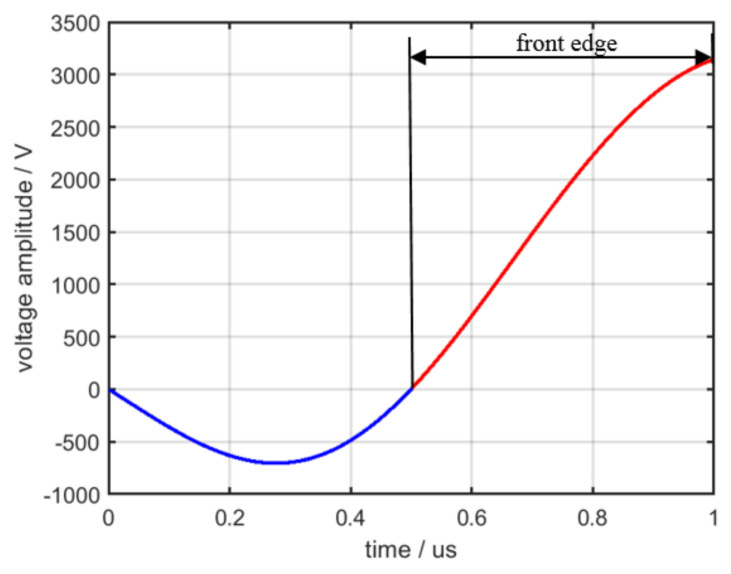
Front edge of the simulation modulation signal.

**Figure 16 sensors-22-02616-f016:**
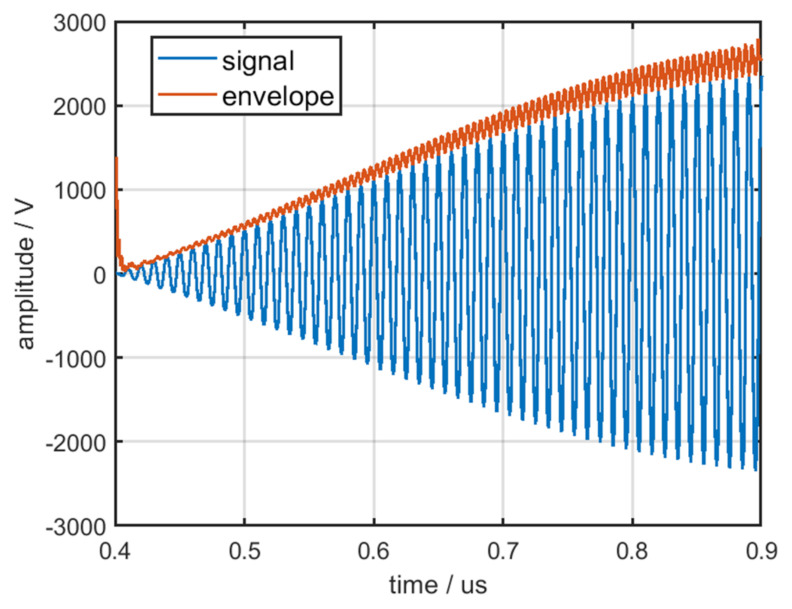
Envelope of the pulse front edge.

**Figure 17 sensors-22-02616-f017:**
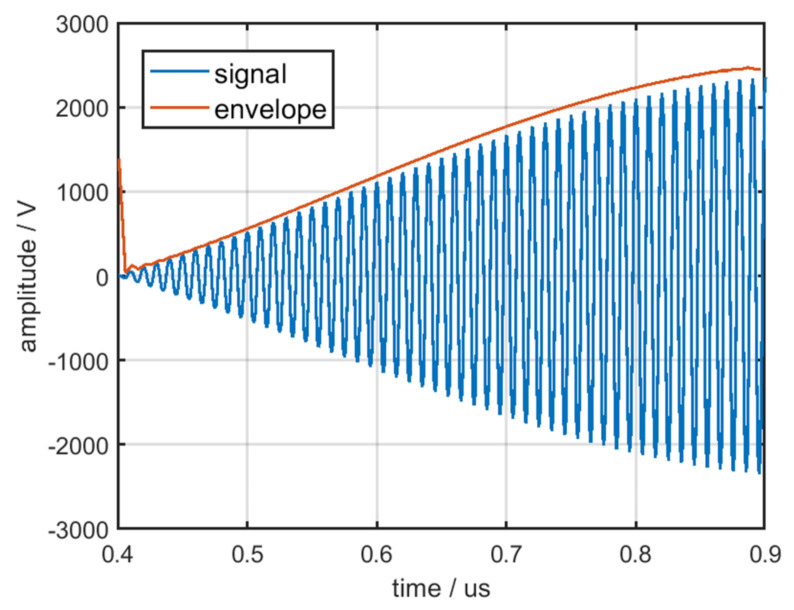
Envelope of the pulse front edge after filtering by downsampling.

**Figure 18 sensors-22-02616-f018:**
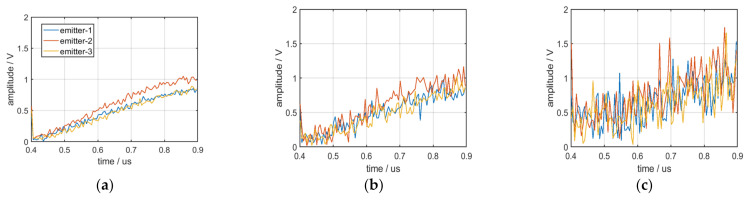
(**a**) Envelope of the pulse front edge at 30 dB SNR; (**b**) envelope of the pulse front edge at 20 dB SNR; (**c**) envelope of the pulse front edge at 10 dB SNR.

**Figure 19 sensors-22-02616-f019:**
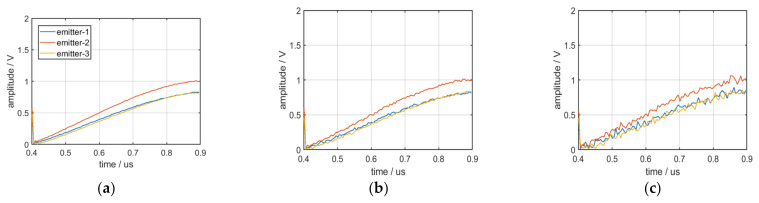
(**a**) Mean curves of the envelope at 30 dB SNR; (**b**) mean curves of the envelope at 20 dB SNR; (**c**) mean curves of the envelope at 10 dB SNR.

**Figure 20 sensors-22-02616-f020:**
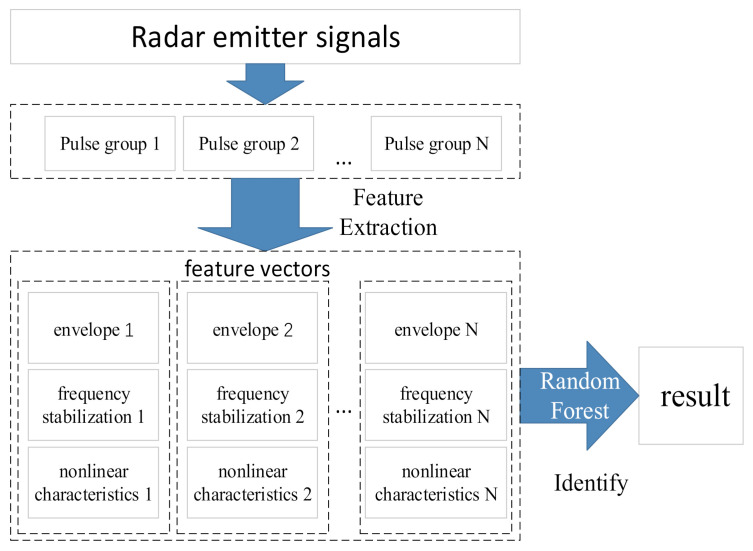
Experimental process.

**Table 1 sensors-22-02616-t001:** Test data of the crystal oscillator frequency stability at 40 °C.

Type	Serial Number	Nominal Frequency	Frequency Stability	Acceptable Value
KOS14D	29	100.000 MHz	0.034 ppm	0.1 ppm
KOS14D	31	100.000 MHz	0.024 ppm	0.1 ppm
KOS14D	33	100.000 MHz	0.030 ppm	0.1 ppm
TCXO14	14	20.000 MHz	0.39 ppm	1 ppm
TCXO14	23	20.000 MHz	0.54 ppm	1 ppm
TCXO14	37	20.000 MHz	0.42 ppm	1 ppm

**Table 2 sensors-22-02616-t002:** Characteristic parameters of 10 simulated emitters.

**Emitter**	**Frequency Stabilization/ppm**	Equivalent Circuit Parameters of Pulse Modulator					Nonlinear Coefficient of RF Amplifier Chain		
		$E_{p}$/v	$Z_{0}$/Ω	$R_{L}$/Ω	$L_{L}/\mathrm{mH}$	$C_{s}/\mathrm{pf}$	$b_{1}$	$b_{2}$	$b_{3}$
1	0.034	12,000	50	10,000	100	900	1.00	0.38	−0.25
2	0.024	12,050	50	10,000	95	920	1.03	0.12	−0.11
3	0.030	12,080	50	9980	105	910	1.01	0.28	−0.22
4	0.038	11,900	50	10,010	102	905	1.01	0.22	−0.18
5	0.031	11,800	51	9990	90	890	1.01	0.30	−0.20
6	0.025	12,100	49	9970	110	880	1.01	0.20	−0.13
7	0.016	11,950	50	10,000	115	895	1.00	0.27	−0.14
8	0.019	12,120	50	10,000	98	915	0.95	0.24	−0.17
9	0.045	12,150	50	10,005	80	905	1.02	0.35	−0.08
10	0.021	11,980	51	10,000	118	885	0.97	0.15	−0.15

**Table 3 sensors-22-02616-t003:** Accuracy with various features.

SNR	All Features	Frequency Stabilization	Nonlinear Coefficients	Envelope of Pulse front Edge
10 dB	92.31%	51.01%	91.82%	84.32%
15 dB	97.58%	50.49%	95.10%	95.58%
20 dB	99.55%	51.03%	98.99%	99.35%
25 dB	99.89%	50.43%	99.71%	99.88%
30 dB	99.94%	50.59%	99.90%	99.90%

**Table 4 sensors-22-02616-t004:** Accuracy with various number of pulses in each group.

SNR	100 Pulses/Group	50 Pulses/Group	20 Pulses/Group	10 Pulses/Group
10 dB	92.31%	93.89%	89.30%	81.51%
15 dB	97.58%	97.98%	96.19%	92.46%
20 dB	99.55%	99.74%	99.29%	98.25%
25 dB	99.89%	99.98%	99.96%	99.88%
30 dB	99.94%	100.00%	100.00%	99.99%

**Table 5 sensors-22-02616-t005:** Accuracy with various methods of specific emitter identification.

SNR	Features Being Extracted + Random Forest	Time-Frequency Graph + LeNet	Envelope + SVM
10 dB	81.51%	64.31%	48.87%
15 dB	92.46%	83.02%	74.05%
20 dB	98.25%	92.59%	90.98%
25 dB	99.88%	97.74%	98.84%
30 dB	99.99%	99.10%	99.99%

## Data Availability

The experimental data in this paper are mainly obtained by mathematical simulation according to the parameters in Table 2.

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
