# Peer review of "Feature Analysis and Extraction for Specific Emitter Identification Based on the Signal Generation Mechanisms of Radar Transmitters"

_sensors, 2022, doi:10.3390/s22072616_

Round 1
Reviewer 1 Report
1. Authors could further compare the performance with wireless channel variations.
2. The caption on the picture is too small
Author Response
Response to Reviewer 1
We are grateful to Reviewer 1 for the thorough review of the manuscript. In response to this review, several changes have been made to the manuscript. These changes seek to address the comments as follows.
Comment1: Authors could further compare the performance with wireless channel variations
Response: Thanks for the reviews’ comment. We noticed that the standard deviation of carrier frequency from the same emitter will change with the carrier frequency, but frequency stabilization will not. Therefore, the frequency stability of carrier is used to identify emitters, and the result of identification will not be affected by the change of carrier frequency.
Comment2: The caption on the picture is too small
Response: Thanks for the reviews’ suggestion. We have modified the figures in our manuscript to make the caption on the figures clearer.
Reviewer 2 Report
Dear authors,
Please consider the following comments before resubmitting your article:
Comment1-You need to modify the conclusion section to emphasize your method’s main achievement compared to other well-known methods.
Comment2-The figures need attention and modification as follow:
- All of the figures need to have a unified style, specifically the linewidth of graphs. And applying boxes around all the plots. I suggest keeping the line width to 1.5 points.
- It is a good practice to keep the scale and axis limits for the data that is compared on different plots, the same if possible. For example in figure 9. Other than considering comment 1 for them, it helps to keep at least b and c the same x-axis limit.
- For figure 10, to show their relation better, you could use a logarithmic scale on the y-axis.
- Figure 13 “composition of pulse modulator” is an elementary diagram that does not give any interesting information to advance readers of Sensor publication. I suggest you improve this diagram.
- It seems that figure 14 is not an original figure generated by the authors, therefore the quality of the figure and the annotations are different and low. I suggest generating your own signal, or finding a suitable reference and citing it accordingly with their permission.
- In figure 6 the time unit should be corrected to its scientific greek letter (in Latex $\mu s$).
- As a suggestion for figure captions provide more explanation so that readers would have a good idea about the figure without referring to your text.
- Make sure you have proper permission from other publishers if you are inserting a figure from another reference.
Author Response
Response to Reviewer 2
We are grateful to Reviewer 2 for the thorough review of the manuscript. In response to this review, several changes have been made to the manuscript. These changes seek to address the comments as follows.
Comment1: You need to modify the conclusion section to emphasize your method’s main achievement compared to other well-known methods.
Response: Thanks for the reviews’ comment. We have modify the conclusion section to emphasize the superiority of our method over other well-known methods.
Comment2: The figures need attention and modification as follow:
- All of the figures need to have a unified style, specifically the linewidth of graphs. And applying boxes around all the plots. I suggest keeping the line width to 1.5 points.
- It is a good practice to keep the scale and axis limits for the data that is compared on different plots, the same if possible. For example in figure 9. Other than considering comment 1 for them, it helps to keep at least b and c the same x-axis limit.
- For figure 10, to show their relation better, you could use a logarithmic scale on the y-axis.
- Figure 13 “composition of pulse modulator” is an elementary diagram that does not give any interesting information to advance readers of Sensor publication. I suggest you improve this diagram.
- It seems that figure 14 is not an original figure generated by the authors, therefore the quality of the figure and the annotations are different and low. I suggest generating your own signal, or finding a suitable reference and citing it accordingly with their permission.
- In figure 6 the time unit should be corrected to its scientific greek letter.
- As a suggestion for figure captions provide more explanation so that readers would have a good idea about the figure without referring to your text.
- Make sure you have proper permission from other publishers if you are inserting a figure from another reference.
Response: Thanks for the reviews’ suggestions. We have modified the figures in our manuscript.
- We have modified the style of all of figures in our manuscript
- Because the content of Figure 13 was not novel enough, we have deleted it.
- We have used the actual signal to modify the Figure 14.
- Figure 5 and Figure 6 were changed to Figure 5 (a) and (b), and the X-axis represents the number of pulses used to calculate the standard deviation of frequency, the y-axis represents standard deviation of frequency
Reviewer 3 Report
1. The author's manuscript said “Figure 1. The composition of MOPA transmitter.”.
The MOPA transmitter gets the nothings.
It is not good to publish the sensors.
2. The author's manuscript said “Figure 3. The composition of PLL frequency synthesizer.”.
The PLL gets the nothings.
It is not good to publish the sensors.
3. The author's manuscript said “3.2. Analysis and extraction of nonlinear characteristics of RF amplifier chain”.
The THD(Total harmonic distortion) gets the same things.
It is not good to publish the sensors.
4. The author's manuscript said “Figure 13. The composition of pulse modulator [17]”.
It is not good to publish the sensors.
5. The author's manuscript said “Figure 15. The equivalent circuit of pulse modulator at the time of pulse front edge[26]”.
It is not good to publish the sensors.
6. The author's manuscript said “3.3. Analysis and extraction of envelope of pulse front edge”
It is good for the Book, and it is not good for the paper.
It is not good to publish the sensors.
7. The author's manuscript said “The results show that the features extracted by this method have "fingerprint characteristics".”
What is the function of the "fingerprint characteristics"?
8. It is not good for QSK, FDM, OFDM, and purpose.
It is not good to publish the sensors.
Round 2
Reviewer 3 Report
1. The envelope of the signal is from small to big signal in Figure 15.
Please show the envelope of the signal from the additional item about big to small signal in Figure 15.
